# New Mechanisms to Prevent Heart Failure with Preserved Ejection Fraction Using Glucagon-like Peptide-1 Receptor Agonism (GLP-1 RA) in Metabolic Syndrome and in Type 2 Diabetes: A Review

**DOI:** 10.3390/ijms25084407

**Published:** 2024-04-17

**Authors:** Jorge E. Jalil, Luigi Gabrielli, María Paz Ocaranza, Paul MacNab, Rodrigo Fernández, Bruno Grassi, Paulina Jofré, Hugo Verdejo, Monica Acevedo, Samuel Cordova, Luis Sanhueza, Douglas Greig

**Affiliations:** 1Pontificia Universidad Católica de Chile, School of Medicine, Division of Cardiovascular Diseases, Santiago 8330055, Chile; lgabrielli@ucchristus.cl (L.G.); paul.mcnab123@gmail.com (P.M.); jrodrigo.fernandezz@gmail.com (R.F.); hugoverdejo@gmail.com (H.V.); macevedo@ucchristus.cl (M.A.); samuelcordovaalvestegui@gmail.com (S.C.); luissanac@gmail.com (L.S.); douglasgreig@gmail.com (D.G.); 2Pontificia Universidad Católica de Chile, School of Medicine, Department of Nutrition and Diabetes, Santiago 8330055, Chile; bagrassi@gmail.com (B.G.); pejofre@uc.cl (P.J.)

**Keywords:** heart failure, HFpEF, obesity, GLP-1, renin angiotensin system, remodeling, AMPK, Rho kinase, aldosterone, microinflammation, diabetes, pericardial adipose tissue

## Abstract

This review examines the impact of obesity on the pathophysiology of heart failure with preserved ejection fraction (HFpEF) and focuses on novel mechanisms for HFpEF prevention using a glucagon-like peptide-1 receptor agonism (GLP-1 RA). Obesity can lead to HFpEF through various mechanisms, including low-grade systemic inflammation, adipocyte dysfunction, accumulation of visceral adipose tissue, and increased pericardial/epicardial adipose tissue (contributing to an increase in myocardial fat content and interstitial fibrosis). Glucagon-like peptide 1 (GLP-1) is an incretin hormone that is released from the enteroendocrine L-cells in the gut. GLP-1 reduces blood glucose levels by stimulating insulin synthesis, suppressing islet α-cell function, and promoting the proliferation and differentiation of β-cells. GLP-1 regulates gastric emptying and appetite, and GLP-1 RA is currently indicated for treating type 2 diabetes (T2D), obesity, and metabolic syndrome (MS). Recent evidence indicates that GLP-1 RA may play a significant role in preventing HFpEF in patients with obesity, MS, or obese T2D. This effect may be due to activating cardioprotective mechanisms (the endogenous counter-regulatory renin angiotensin system and the AMPK/mTOR pathway) and by inhibiting deleterious remodeling mechanisms (the PKA/RhoA/ROCK pathway, aldosterone levels, and microinflammation). However, there is still a need for further research to validate the impact of these mechanisms on humans.

## 1. Obesity and Heart Failure with Preserved Ejection Fraction (HFpEF)

Heart failure with preserved ejection fraction (HFpEF) is a growing epidemic, accounting for 50% of all heart failure patients, and is the leading cause of hospitalizations in patients over 65 years of age [1]. Increasing prevalence, morbidity, and clinical inertia have encouraged a reconsideration of the pathophysiology of HFpEF over the last few years (Table 1). Compared with heart failure with reduced ejection fraction, HFpEF has distinct clinical phenotypes [1] and a burden of major comorbidities (Table 1).

Possibly, the keystone of heart failure (HF) is left ventricular (LV) remodeling. Without LV remodeling, there is no HF. In heart failure with reduced ejection fraction (HFrEF), systolic dysfunction leads to eccentric hypertrophy and myocardial fibrosis. In HFpEF, progressive LV wall thickening leads to concentric hypertrophy (LVH) with impaired myocardial relaxation and increased stiffness, causing LV diastolic dysfunction and HFpEF. Comorbidities such as age, hypertension, obesity, type 2 diabetes (T2D), and renal dysfunction (as shown in Table 1) strongly contribute to LV concentric remodeling.

Distinct clinical HFpEF phenotypes are recognized. There are six major phenotypes of HFpEF described, characterized by distinct clinical features: the aging phenotype, the obesity or cardiometabolic phenotype, the phenotype associated with arterial hypertension, the pulmonary arterial hypertension phenotype, the coronary artery disease phenotype, and the phenotype associated with left atrial myopathy [3]. All phenotypes of HFpEF have a variety of comorbidities in common. It is uncommon for HFpEF patients to have only one comorbid condition. The aging phenotype is often associated with comorbidities such as atrial fibrillation, anemia, chronic obstructive pulmonary disease (COPD), and frailty. On the other hand, the obesity phenotype is more commonly associated with comorbidities such as OSA, diabetes, and chronic kidney disease (CKD) [4]. Phenotyping HFpEF allows for tailoring therapeutic modalities for concentric LV remodeling reversal and, eventually, better outcomes. The obese–diabetic phenotype of HFpEF is extremely common [4] and associated with poor outcomes [5]. Obesity is the main driver of T2D, with 90–95% of patients with T2D being obese [6]. Obesity and T2D overlap in the development and progression of HFpEF [7]. The obese–diabetic–hypertensive phenotype (cardio–metabolic phenotype) is the most common phenotype found in clinical practice and carries the greatest burden of morbidity and mortality [1], and cardiac remodeling plays a major role in its pathophysiology. Obesity, metabolic syndrome, T2D, and hypertension are highly prevalent all over the world. The World Health Organization reports that 1.28 billion adults aged 30–79 worldwide have hypertension [8]. In 2016, 39% of adults aged 18 and over were overweight, and 13% were obese [8]. Additionally, approximately 422 million people worldwide have diabetes [8]. The prevalence of metabolic syndrome varies globally, ranging from 12.5% to 31.4%, depending on the definition used [9].

The prevalence of these four conditions depends on age [10,11,12,13]. As we age, both overweight and obesity become more common, reaching their highest point between the ages of 50 and 65 years, and then showing a slight downward trend [11]. It is worth noting that the prevalence of hypertension in adults in the United States (2017–2018) increases with age: 22.4% (aged 18–39), 54.5% (40–59), and 74.5% (60 and over) [10]. These high prevalences are associated with a high incidence of HFpEF [14,15,16]. To effectively prevent this, a recent randomized trial with GLP-1 RA in already obese HFpEF patients suggests taking action. The trial demonstrated larger reductions in symptoms and physical limitations, greater improvements in exercise function, and more significant weight loss than a placebo [17].

This review analyzes how obesity affects HFpEF pathophysiology and discusses novel mechanisms for the prevention of HFpEF using a glucagon-like peptide-1 receptor agonism (GLP-1 RA) in patients with obesity, with metabolic syndrome, or in obese T2D patients, specifically addressing the consequences of obesity on the heart and on HFpEF, the counter-regulatory renin angiotensin system, and prevention of cardiovascular damage in this context. Finally, the GLP-1 axis is addressed as an anti-diabetic, anti-obesity, and anti-cardiac remodeling pathway.

## 2. Consequences of Obesity on the Heart and on Heart Failure with Preserved Ejection Fraction

The regional distribution of adipose tissue and the development of ectopic fat are major determinants of metabolic and cardiovascular diseases [18]. Epicardial adipose tissue (EAT) is an ectopic fat depot located between the myocardium and the visceral pericardium in close contact with coronary vessels. EAT is an extremely active endocrine organ with a high capacity for releasing and taking up free fatty acids [18]. EAT secretes inflammatory cytokines, releases excessive fatty acids, and increases the mechanical load on the myocardium, resulting in myocardial remodeling [19]. EAT can be both highly protective for the adjacent myocardium through its dynamic brown fat-like thermogenic function and deeply harmful via paracrine or vasocrine secretion of pro-inflammatory and profibrotic cytokines [20]. EAT is a major source of adipokines, chemokines, and cytokines, interacting paracrinally or vasocrinally with vascular cells or myocytes. The expression and secretion of pro-inflammatory cytokines, including IL-6, IL-1β, MCP-1, and TNF-α, are higher in EAT than in subcutaneous fat [8]. This is due in part to the upregulation of nuclear factor-kB (NF-kB) and c-Jun N-terminal kinase (JNK) [21]. EAT modulates the metabolic environment of the coronary arteries and myocardium and is associated with various health conditions, including coronary artery disease [19,20], metabolic syndrome [21,22], type 2 diabetes (T2D) [23,24], chronic obstructive pulmonary disease [25], obstructive sleep apnea [26], aging [27], and male sex [28]. These conditions are also risk factors for poor COVID-19 prognosis, and the infection could trigger a systemic inflammatory response [29,30]. A recent meta-analysis found that patients in the severe COVID-19 group had higher EAT measures compared with the non-severe group [30]. Therefore, there is a probability that EAT could transmit this inflammation to the heart.

Each metabolic disorder that is linked to both atrial fibrillation (AF) and HFpEF is also accompanied by an expansion of epicardial adipose tissue (EAT) mass [31]. Obese individuals are at a significantly higher risk of developing both AF and HFpEF.

Leptin and adiponectin are two adipokines that can modulate insulin sensitivity and have been shown to have pro-inflammatory and anti-inflammatory effects, respectively [15]. Obesity is associated with increased leptin levels that activate the cells of the innate and adaptive immune system [32]. Leptin acts as an acute-phase reactant, increasing the secretion of proinflammatory cytokines such as IL-6, IL-12, and TNF-α in macrophages [33]. This, in turn, increases leptin expression in adipose tissue and circulating leptin, creating a feedback loop that promotes the inflammatory state. Obese individuals exhibit a greater abundance of macrophages in their adipose tissue, indicating that both adipocytes and macrophages play significant roles in inflammation in obesity [33]. High levels of leptin in obesity are associated with cardiac and renal fibrosis [34], increased aldosterone production, and sodium retention [35].

On the other hand, adiponectin, which is exclusively produced by adipocytes and circulates in plasma, has a significant insulin-sensitizing effect in humans [16] and exerts anti-inflammatory effects, especially in cardiometabolic diseases [36]. Serum adiponectin levels are inversely correlated with obesity and inflammatory cytokine levels in T2D, and proinflammatory cytokines reduce adiponectin expression in adipocytes [36]. In vitro treatment of macrophages with adiponectin inhibits the pro-inflammatory cytokine TNF-α and induces the expression of the anti-inflammatory cytokine IL-10 [37]. In addition, adiponectin induces the M1 to M2 macrophage polarization switch, thereby attenuating chronic inflammation [36]. Low adiponectin levels in obesity increase the risk for cardiovascular disease [35], whereas adiponectin levels are reduced in HFpEF. In vitro, adiponectin has several protective effects: it stimulates AMP-activated protein kinase (AMPK)-dependent and extracellular signal-regulated kinase signaling in cardiac myocytes and endothelial cells [38], reduces hypertrophy and fibrosis, activates the endothelial nitric oxide synthase system, and increases nitric oxide production [39,40].

In young adults with obesity, plasma leptin levels were 2.6 times higher and GLP-1 levels were 90% higher compared to those with a lean body type. In addition, the Homeostatic Model Assessment for Insulin Resistance (HOMA) index, which is an estimator of insulin resistance, was five times higher in this population [41]. Obese subjects had 100% higher plasma leptin levels, 2.7-fold higher HOMA levels (*p* < 0.001), and 20% lower plasma adiponectin levels (*p* = 0.003) compared to their lean counterparts [42]. Obesity treatment can cause changes in these hormone levels. After bariatric surgery, a new systematic review found that leptin concentrations decreased following most bariatric surgeries [43]. This might be due to the correction of leptin resistance, which is increased in obesity [43]. On the other hand, adiponectin has been shown to prevent obesity and increase during weight loss, and in the abovementioned systematic review, this was observed in most surgeries [43].

Observational and community-based cohorts have reported an association between obesity and LV concentric remodeling [44,45]. Evidence of the central role of obesity in the pathogenesis of LV concentric remodeling is further supported by the correlation between weight loss and reduction of LV mass [46,47], and also between weight loss, LVH regression, and blood pressure reduction after bariatric surgery [48,49,50]. Obesity can impair LV diastolic function through mechanisms other than increased cardiac pre- and after-load [1] and is a known risk factor for incident HFpEF [34,51,52].

Interestingly, HFpEF correlates more closely with visceral adipose tissue (VAT) mass than with body mass index [53,54], and peak aerobic capacity is inversely and independently related to intra-abdominal fat. VAT predicts hospitalizations in HFpEF patients [55,56,57]. In the I-PRESERVE trial in HFpEF, 71% of the patients had a BMI > 26.5 kg/m^2^ [58]. In the RELAX trial in HFpEF, 55% of the patients had a BMI > 35 kg/m^2^ [59]. In five recent clinical trials in HFpEF, the prevalence of BMI > 30 kg/m^2^ was between 34 and 64.8% [58,59,60,61,62]. Obesity contributes to the development of HFpEF by increasing renal tubular sodium reabsorption and expanding plasma volume [1] with overproduction of aldosterone through two pathways: (1) renin angiotensin system (RAS) activation and increased aldosterone secretion from the adrenal cortex and adipocytes and (2) direct leptin stimulation of adrenal cortical cells [63]. Natriuretic peptides have been shown to reduce aldosterone levels. However, it is important to note that in cases of obesity, neprilysin activity, which metabolizes natriuretic peptides, is higher. This may potentially limit the effectiveness of natriuretic peptides in reducing increased aldosterone secretion [64]. Hyperaldosteronism also stimulates the accumulation and inflammation of epicardial adipose tissue leading to increased local and systemic inflammation [34].

### 2.1. Obesity, Low-Grade Systemic Inflammation, Cardiac Damage, and HFpEF

Adipocyte homeostasis is maintained through the modulation of pro-inflammatory and anti-inflammatory cytokines. Obesity promotes increased pro-inflammatory cytokines and dysregulation of adipokines. Adipose tissue exerts an endocrine effect through adipokines, and obesity-induced adipocyte dysfunction modifies adipokine levels [65], promoting left ventricular remodeling and heart failure with preserved ejection fraction (HFpEF) [66].

Adipose tissue (AT) is stored in various compartments, with the subcutaneous compartment accounting for approximately 80% and the remaining 20% distributed among visceral and other compartments, such as epicardium, perivascular, hepatic, pancreas, renal, and skeletal muscle [67]. Visceral adipose tissue (VAT) refers to the accumulation of fat within the abdominal cavity, including the mesentery and omentum. It can be measured using single-slice CT or multiple-slice MRI imaging [68]. Weight gain leads to accumulation of AT through adipocyte hypertrophy or hyperplasia. Visceral AT expansion leads to dysfunction and inflammation, which, in turn, stimulates low-grade systemic inflammation [69]. An increase in BMI is associated with elevated levels of circulating inflammatory markers such as C-reactive protein, interleukin-6, P-selectin, vascular cell adhesion molecule 1, plasminogen activator inhibitor 1, and tumor necrosis factor-alpha (TNF-α) [70,71].

Following adipocyte hypertrophy or hyperplasia, visceral adipose tissue undergoes a transition from an anti-inflammatory state that facilitates angiogenesis and lipid storage to a pro-inflammatory state. This transition is marked by the release of monocyte chemoattractant protein 1, C-X-C motif chemokine 12, leukotriene B4, and colony-stimulating factor 1, which promote proliferation of activated macrophages and infiltration of adipose tissue by macrophages [72,73]. In addition, when nutrient availability is excessive, adipocytes store various lipid species in lipid droplets that rapidly increase in size and reach the diffusion limit of oxygen [74]. Thus, hypoxia is an important driver of the early AT response to increased dietary lipids [74]. Adipocyte hypoxia is induced as early as 1 and 3 days during a high-fat diet [75]. Hypoxia during early AT expansion induces stress signaling that facilitates angiogenesis, initiates inflammatory cell infiltration, and induces organized ECM remodeling to encourage appropriate AT expansion [74]. Adipogenesis affects the process of adipose tissue remodeling, while hypoxia triggers angiogenesis, extracellular matrix remodeling, and inflammation [74,75,76].

Inflammatory visceral adipose tissue produces reactive oxygen species and low levels of nitric oxide, leading to mitochondrial dysfunction and activation of the Nod-like receptor protein 3 inflammasome [77], which serves as a platform for caspase-1 activation leading to the processing of proinflammatory cytokines IL-1β, IL-18, and gasdermin D (GSDMD)-mediated cell death [78]. This form of cell death represents a major pathway of inflammation [78]. Low-grade systemic inflammation worsens cardiovascular function [79,80,81]. It heightens an inflammatory response in the coronary microvasculature [82], which modifies cardiomyocyte elasticity and function, increases myocardial collagen deposition, impairs myocardial relaxation, and results in diastolic dysfunction and HFpEF [83]. Microvascular inflammation with macrophages and secretion of TGF beta results in LV deposition of high-tensile collagen [84]. Increased myocardial collagen (myocardial fibrosis) increases left ventricular stiffness and filling pressure, leading to diastolic dysfunction. This makes passive inflow difficult and causes backward diastolic flow, further impairing early filling [85]. In this regard, there is a direct relationship between the level of myocardial collagen, LV stiffness, and pulmonary capillary wedge pressure [86]. Additionally, microvascular rarefaction and defects in the endothelial cell metabolic programming and angiogenesis, which are dependent on Sirtuin 3, may affect the progression of perivascular and myocardial fibrosis in HFpEF as well [87].

### 2.2. Visceral Adipose Tissue and HFpEF

The accumulation of visceral adipose tissue (VAT) plays a significant role in the development and progression of cardiometabolic conditions. In type 2 diabetes (T2D), visceral AT is a strong predictor of insulin resistance [88] and increased cardiometabolic risk [89]. The body’s inability to manage unrestricted energy intake leads to the expansion of visceral AT, which mediates most of the negative impact of obesity on clinical outcomes. The Multi-Ethnic Study of Atherosclerosis (MESA) [56] found that patients with increased visceral adipose tissue had a 2.2-fold higher risk of incident hospitalization due to heart failure with preserved ejection fraction (HFpEF). Incident HFpEF hospitalizations in the population of the Jackson Heart Study were significantly associated with both visceral and epicardial AT, but not with subcutaneous AT [90]. The only significant variable that predicted all-cause mortality was epicardial AT. There was a trend towards increased all-cause mortality with increased visceral AT, but no significant trend was seen with subcutaneous AT [90].

Visceral AT accumulation is associated with left ventricular (LV) diastolic dysfunction, increased LV mass, sphericity, and lower-end diastolic volumes in patients with obesity and HFpEF [54,91]. Additionally, the effects of visceral AT are gender-specific, with women having a higher percentage of visceral AT at the baseline and worse hemodynamics in HFpEF [92]. Women with increased visceral AT and HFpEF also exhibit higher exercise-induced LV-filling pressures [92].

### 2.3. The Pericardial/Epicardial Adipose Tissue and HFpEF

Obesity and T2D are independently associated with increased pericardial/epicardial AT [93]. Epicardial adipose tissue (EAT), a component of VAT located in the heart, has metabolic, mechanical, and thermogenic roles [19]. EAT is twice as metabolically active as normal white AT and plays a significant role in lipolysis and free fatty acid release [94]. This leads to high levels of circulating free fatty acids, resulting in increased deposition of cardiac triglycerides. As epicardial adipose tissue lies directly on the myocardium, any free fatty acids released by it may have a direct effect on myocytes and coronaries due to the lack of a fibrous fascial layer between the two. A large release of free fatty acids may lead to cardiac lipotoxicity [29]. More interestingly, in a study on 33 patients with cardiovascular diseases undergoing open heart surgery aimed to assess the expression of GLP-1R as well as the glucose-dependent insulinotropic polypeptide receptor (GIPR) and the glucagon receptor (GCGR), it was found that human EAT expresses GLP-1R, GIPR, and GCGR at similar levels of mRNA and protein in macrophages and partially in adipocytes [95], suggesting that these G-protein-coupled receptors could be targeted by multi-receptor innovative drugs with cardiometabolic benefits beyond their effects on glucose and body weight [95,96].

Patients with increased epicardial adipose tissue have a high LV mass index, large left atrial size, and high E/e’ velocity under echocardiography [97,98,99], which precedes HFpEF. The association between epicardial AT and LV functional parameters remains significant even after adjusting for obesity markers and traditional cardiovascular risk factors [100]. EAT may lead to an increase in myocardial fat content and interstitial fibrosis, which can reduce myocardial diastolic function and contractility, as evidenced by reduced global longitudinal strain [101]. In patients with substantial obesity with cardiomegaly and increased epicardial fat volume, the pericardial sac is stretched to a steeper pressure–volume relationship and there is enhanced diastolic ventricular interaction and pericardial restraint [102]. This exerts an exaggerated compressive force on the heart, resulting in higher left- and right-sided filling pressures for any degree of left ventricular (LV) end diastolic volume—the true LV preload [103]. In a recent study of HFpEF in 28 patients (17 women, BMI 37 ± 7 kg/m^2^, aged 70 ± 6 years), epicardial fat was strongly related to the LV eccentricity index, which is evidence of an adverse cardio–mechanical interaction [103]. In this study, the LV eccentricity was not related to body mass index, subcutaneous fat, visceral fat, or pulmonary vascular resistance and exercise did not modify the cardio–mechanical interaction [103]. Patients with diastolic dysfunction and increased EAT show more pronounced signs of diastolic functional failure and adverse structural remodeling [104]. Despite similar morphological characteristics, patients with high EAT show significant cardiac functional impairment, in particular in the atria, indicating that regionally increased EAT directly induces atrial functional failure [104] (Figure 1).

Recently, it has been shown that in HFpEF patients obesity and epicardial adiposity are associated with hemodynamic signs of pericardial constraint [106], and 73% of them had a square root sign [106]. Increased epicardial AT also results in lower peak VO_2_ consumption and peripheral O_2_ extraction in patients with HFpEF, indicating a worse hemodynamic profile [107]. In the MESA study, it was found that increased epicardial AT volumes were correlated with a two-fold higher incidence of HFpEF in women and 53% in men (*p* < 0.001) without correlation with higher HFrEF incidence [108]. In samples of epicardial adipose from patients with HFpEF, differentially expressed circular RNAs corresponded to genes mainly involved in the regulation of cellular and metabolic processes [109].

### 2.4. The Relationship between Type 2 Diabetes, Obesity, and HFpEF

In addition to T2D macrovascular complications, T2D has direct effects on the myocardium, such as left ventricular hypertrophy [110,111], a complex myocardial dysfunction referred to as diabetic cardiomyopathy (which results in abnormal diastolic and systolic function) [112,113], heart failure [114,115,116], and also prolonged ventricular repolarization [117].

Patients with T2D and HFpEF exhibit increased t-tubule density and collagen deposition compared to HFrEF patients. They also have impaired diastolic calcium homeostasis, including lower sarco/endoplasmic reticulum Ca^2+^-ATPase activity, indicating a different pathophysiological process when compared to non-T2D HFpEF [118]. HFpEF patients with type 2 diabetes often have higher BMIs than non-diabetic HFpEF patients [59,119]. They also tend to have more severe initial presentations, more hospitalizations, more left ventricular hypertrophy (LVH), higher filling pressures, and a trend towards higher LV mass and increased cardiac fibrosis, as observed by cardiac magnetic resonance imaging [120].

Exercise intolerance is a common feature of both T2D and HFpEF [121] due to the impairment of cardiac performance [121] and skeletal muscle metabolism/perfusion [122]. In HFpEF, obesity and T2D significantly contribute to exercise intolerance. Obesity-induced sarcopenia increases muscle mass loss due to aging and exercise intolerance [123]. T2D reduces exercise capacity through impaired cardiac energetics and skeletal muscle oxygen extraction and metabolism [124,125]. Exercise intolerance can lead to poor quality of life, frequent re-hospitalizations, and early mortality in patients with T2D and HFpEF [125]. Therefore, reversing exercise intolerance is an important therapeutic target for patients with T2D–HFpEF. Obesity and T2D are additive risk factors in patients with T2D–HFpEF. Obesity directly affects the severity of T2D and HFpEF, and worsens outcomes in T2D. Insulin resistance results in increased insulin production from pancreatic β-cells, which eventually cannot meet glycemic demands. Ectopic deposition of pancreatic fat also contributes to β-cell dysfunction and, in turn, to the development of type 2 diabetes [126]. Therefore, the treatment of the obese HFpEF phenotype should focus on addressing obesity and T2D.

### 2.5. Current Effective Therapeutic Interventions in Patients with the Obese–HFpEF Phenotype

A systematic review and meta-analysis have provided evidence that exercise, diet, bariatric surgery, and pharmaceutical interventions can reduce cardiac adipose tissue volume [127]. A significant pooled effect size for reduction in the EAT volume was observed following weight-loss interventions as compared with control interventions (ES = −0.89, 95% CI: −1.23 to −0.55, *p* < 0.001). Significant weight loss can reduce epicardial adiposity and improve the development of both atrial fibrillation (AF) as well as HFpEF [31]. A recent meta-analysis has shown that EAT surrounding the left atria quantified by computed tomography seems to be predictive of post-ablation AF recurrence [128]. Studies have shown that weight loss has a positive impact on the clinical course of HF, particularly in patients with HFpEF. These patients experience improved diastolic filling dynamics, symptoms, and exercise tolerance [31]. In a 6-month randomized, open-label, controlled study, 95 T2D subjects (BMI ≥ 27 kg/m^2^, hemoglobin A1c 8%) on metformin monotherapy were randomized to receive the GLP-1 RA liraglutide or to remain on metformin. The thickness of EAT was significantly reduced from 9.6 to 6.8 and 6.26 mm at 3 and 6 months, respectively, with the use of liraglutide [129]. In contrast, no reduction in EAT was observed in the metformin group. Additionally, the liraglutide group showed improvements in BMI and hemoglobin A1c [129]. In a recent clinical study, 62 patients underwent cardiac magnetic resonance (CMR) before and after bariatric surgery. On average, the patients showed 38.9% excess weight loss at 212 days and 64.7% at 1030 days following bariatric surgery. Most abdominal visceral adipose tissue and EAT loss (43% and 14%, respectively *p* < 0.0001) occurred within the first 212 days, with non-significant reductions thereafter [105]. Currently, bariatric surgery is the most effective weight loss therapy available and also has independent benefits for epicardial fat deposits, which may have advantages in terms of relieving pericardial restraint [102].

Recently, positive results have emerged regarding the effectiveness of therapies for HFrEF in lowering the mortality rate for HFpEF. In a TOPCAT trial, patients with obesity and T2D benefited the most from the mineralocorticoid receptor blocker spironolactone [130]. The primary endpoint, which included all-cause death and hospitalization due to heart failure, was observed to be maximally reduced in patients with a BMI greater than 33 kg/m^2^ [131]. The study showed that spironolactone was more effective in patients with high waist circumference, indicating increased visceral AT. Additionally, six studies involving 5201 participants found that sacubitril/valsartan reduced heart failure hospitalization rates compared to ACEIs and ARBs (relative risk, 0.78; 95% CI, 0.65 to 0.85; *p* = 0.001) [132]. Two recent large randomized clinical trials have demonstrated the benefits of SGLT2 inhibitors [62,133] compared to a placebo in patients with HFpEF and NYHA HF classes II to IV symptoms, structural heart disease (left atrial enlargement or left ventricular hypertrophy), recent HF hospitalization, elevated BNP levels, and an EF greater than 40% [134]. Both studies showed that SGLT2 inhibitors resulted in an 18% to 21% reduction in the rate of hospitalization due to heart failure or cardiovascular death [134].

## 3. The Counter-Regulatory Renin Angiotensin System and Prevention of Cardiovascular Damage

Chronic activation of the “classic” renin–angiotensin system (RAS) promotes cardiovascular damage, an effect that is antagonized by components of the non-canonical or counter-regulatory RAS. This counter-regulatory RAS axis consists of angiotensin 1–7 (Ang 1–7), angiotensin 1–9 (Ang 1–9), angiotensin-converting enzyme 2 (ACE2), type 2 angiotensin II receptor (AT2R), proto-oncogene Mas receptor, and Mas-related G protein-coupled receptor member D [135]. Components of the counter-regulatory RAS, including Ang 1–7, Ang 1–9, alamandine, and their receptors, are protective in several cardiovascular diseases, such as hypertension and heart failure [135] (Figure 2).

In the classical RAS, renin cleaves angiotensinogen to produce Ang I, which can be hydrolyzed with angiotensin-converting enzyme (ACE) to Ang II, which binds to the Ang II receptors type 1 (AT1R) and 2 (AT2R). AT1R activation increases aldosterone and anti-diuretic hormone production, sympathetic nervous system tone, BP, vasoconstriction, cardiac hypertrophy, fibrosis, inflammation, vascular smooth muscle cell dedifferentiation, and reactive oxygen species production, and at the same time, reduces parasympathetic nervous system (PSNS) tone, baroreflex sensitivity, nitric oxide production, and natriuresis [135]. Ang I can also be cleaved by ACE2 and neprilysin (NEP) to produce Ang 1–9 and Ang 1–7. Ang 1–9 activates the AT2R to trigger natriuresis and NO production, promoting vasodilatory effects and reducing blood pressure (BP). Ang 1–9 is cardioprotective and attenuates inflammation, cardiac hypertrophy, and fibrosis [85]. Ang 1–7 binds to the Mas receptor (MasR) and reduces BP and noradrenaline release in hypertensive rodents [86]. Activation of the MasR increases NO generation, natriuresis, vasodilatation, PSNS tone, and baroreflex sensitivity. Ang 1–7 is also formed from Ang II cleavage by ACE2 and can be further metabolized to alamandine [135].

Several preclinical studies have shown the advantageous effects of the counter-regulatory RAS. However, clinical studies confirming these observations are still limited. It has been observed that prehypertensive subjects have higher ACE2 activity compared to hypertensive subjects [136]. In patients with heart failure (HF), ACE2 activity is strongly correlated with a clinical diagnosis of HF, worsening LV ejection fraction, increased brain natriuretic peptide (BNP) levels, and the severity of HF [137].

The counter-regulatory RAS exhibits an inverse relationship between Ang II with Ang 1–9 and ACE2 levels [135]. However, there is limited human data available on the levels of cardioprotective Ang 1–9 and ACE2 in MS, T2D, and heart failure. Plasma Ang II levels were 29% higher in stable T2D patients compared to the controls (<0.05) [138]. In a mechanistic study in patients with heart failure, plasma Ang 1–9 levels were 86% lower and Ang II levels were 30% higher compared to the controls (*p* < 0.05) [139]. Currently, there are no additional data on Ang 1–9 levels in humans. In a small study of 11 patients with CKD and T2D, plasma levels of Ang 1–7 increased by 2.3-fold after 12 weeks of empagliflozin treatment [140]. However, there are no reports on the effect of any GLP-1 RAs on Ang 1–7, 1–9, ACE 2, or aldosterone levels in patients with MS or T2D in relation to cardiac phenotype protection.

A meta-analysis conducted recently found that lower adipose tissue ACE2 expression in humans, specifically in abdominal subcutaneous fat, was associated with multiple adverse cardio–metabolic health indices. These included T2D, obesity status, increased serum fasting insulin levels and BMI, and lower serum HDL levels [141]. Specifically, individuals with cardio–metabolic features were found to have lower ACE2 expression and a reduced proportion of microvascular endothelial cells but a high proportion of macrophages [141]. Thus, decreased ACE2 expression in adipose tissue may play a role in the development of cardio–metabolic disorders as well as in the increased risk of severe COVID-19.

In obese adolescents, Ang (1–7) levels correlated inversely with weight, BMI, leptin, and diastolic and systolic blood pressure [142], as well as with skinfolds, waist–hip ratio, leptin, and arm circumference. Plasma Ang I levels were higher in obese groups compared to the normal weight group, but plasma Ang II levels were similar in all groups [142], suggesting that Ang-(1–7) is a novel biomarker of childhood obesity.

## 4. The Glucagon-like Peptide 1 (GLP-1) Axis: A Novel Anti Diabetic, Anti-Obesity, and Anti Myocardial Remodeling Path

Glucagon-like peptide 1 (GLP-1) is a glucagon incretin hormone released from the gut enteroendocrine L-cells. GLP-1 (and GLP-2) are continuously secreted from enteroendocrine cells at low basal levels in the fasting or interprandial state. Thus, circulating levels of these peptides increase rapidly within minutes of food ingestion [143]. GLP-1 controls glycemic variation during meals by increasing insulin secretion and decreasing glucagon secretion. Additionally, it slows gastric emptying and reduces food intake, which maximizes nutrient absorption while limiting weight gain. These effects are mediated through the GLP-1 receptor (GLP-1 R). GLP-1 Rs were first identified in islet β cells and in the central nervous system. They were subsequently found in islet and pancreatic exocrine cells, as well as in the autonomic and enteric nervous systems, blood vessels, Brunner’s glands, and the sinoatrial node [143]. Currently, GLP-1 is a therapeutic drug for type 2 diabetes mellitus, MS, and obesity [144]. It has a short half-life of less than 2 min in vivo, is degraded by dipeptidyl peptidase 4 (DPP4) [145], and is eliminated via the kidneys [146]. GLP-1 reduces blood glucose levels by stimulating insulin synthesis, suppressing islet α-cell function, and promoting the proliferation and differentiation of β-cells while reducing apoptosis [147,148]. Additionally, GLP-1 and its agonists have been shown to promote cardiovascular protection after myocardial infarction, congestive heart failure, and ischemia, as well as have protective effects on various cardiovascular diseases and conditions, such as hypertension, atherosclerosis, and myocardial hypertrophy [148].

### 4.1. Effects of Glucagon-like Peptide 1 (GLP-1) upon Glucose Regulation, Obesity, and Diabetes

Since in obese HFpEF, visceral and epicardial AT drive LV remodeling, aggressive weight management will benefit HFpEF patients with obesity or T2D [1]. Currently, there are five FDA-approved medications for weight loss, namely phentermine-topiramate, orlistat, naltrexone-bupropion, and the two GLP-1 RAs liraglutide and semaglutide [149]. All GLP-1 RAs stimulate insulin and suppress glucagon secretion, but they differ in structure, resulting in differences in duration of action, dosing, formulation, side effects, glucose lowering, weight loss, and compliance [150]. GLP-1 RAs are administered subcutaneously (except oral semaglutide). Short-acting GLP-1 RAs tend to have a more pronounced effect on postprandial hyperglycemia and gastric emptying and less of an effect on fasting glucose than longer-acting GLP-1 RAs. Longer-acting agents, such as liraglutide and oral semaglutide, are administered once daily, whereas dulaglutide, exenatide extended release, and subcutaneous semaglutide are given once weekly. Glucose-lowering efficacy may be highest for subcutaneous semaglutide followed by dulaglutide, liraglutide, and exenatide [150].

### 4.2. Effects of GLP-1R Agonism on the RAS, Cardiac Remodeling Pathways, and HFpEF Mechanisms

There are some observations on the relationship between the GLP-1R agonism and the RAS. GLP-1R activation has the potential to interact with RAS activity [151]. In mice, GLP-1R activation by exendin-4 reduced intrarenal RAS activity and the Ang II-mediated TGF-β1/Smad3 signaling pathway [152]. In spontaneously hypertensive rats (SHR), both GLP-1 RAs, liraglutide and alogliptin, significantly reduced circulating levels of Ang II and upregulated levels of cardiac AT2R and cardiac ACE2 [153]. In a study of 12 healthy young men, GLP-1 infusion using a synthetic human GLP-1 for 2 h reduced Ang II levels by 19% (*p* = 0.003) and induced natriuresis [153]. These preclinical and clinical observations suggest that GLP-1 inhibits ACE and activates ACE2.

Pathological myocardial hypertrophy is a fundamental process in heart failure. In hypertensive mice induced by angiotensin II, GLP-1R activation normalized blood pressure and reduced cardiac hypertrophy, vascular fibrosis, endothelial dysfunction, oxidative stress, and vascular inflammation [154]. In spontaneously hypertensive rats (SHRs), the GLP-1R agonist liraglutide reduces heart weight and cardiac muscle cell volume [155]. In cultured H9C2 cardiac cells stimulated by Ang II, GLP-1 RA treatment reduces myocardial cell volume; inhibits the expressions of atrial natriuretic peptides, β-myosin heavy chain, and Rho kinase (ROCK) 2; and reduces myosin light chain and Myosin phosphatase target subunit 1 (MYPT1) phosphorylation, a marker of increased ROCK activity [155]. Treatment of H9C2 cardiac cells with a protein kinase A (PKA) inhibitor resulted in the disappearance of the effect of GLP-1, while the inhibitory effect was increased with the ROCK inhibitor Y-27632. These findings suggest that GLP-1 may reverse myocardial hypertrophy through the PKA/RhoA/ROCK signaling pathway [155], as shown in Figure 3A. In mice fed with the normal chow diet and given the ROCK inhibitor Y-27632, increased numbers of L cells in intestinal tissues, increased plasma levels of GLP-1 and insulin, and lower blood levels of glucose were observed compared with the control mice [156]. In diabetic mice (low streptozotocin doses and high-fat diet), the GLP-1 analog exendin-4 administered during 8 weeks significantly reversed the increased myocardial lipid accumulation, oxidative stress, apoptosis, and cardiac remodeling and dysfunction [157]. Exendin-4 inhibited abnormal activation of the (PPARα)-CD36 pathway by stimulating protein kinase A and suppressing the ROCK pathway in DM hearts [157].

In SHRs, both GLP-1 RAs liraglutide and alogliptin were effective in reducing cardiac hypertrophic markers such as atrial and brain natriuretic peptides and β-myosin heavy chain, as well as blood pressure and associated histological changes [158]. Additionally, both GLP-1 RAs were found to significantly reduce Ang II and AT1R levels while upregulating ATR2 and ACE2 levels. This was indicated by a lower AT1R/AT2R ratio [158]. Liraglutide and alogliptin significantly increase GLP-1 receptor expression and adenosine monophosphate-activated protein kinase (AMPK) phosphorylation while reducing the phosphorylation of mammalian target of rapamycin (mTOR) and p70 ribosomal S6 protein kinase [158]. In the study mentioned above, inhibition of protein synthesis by AMPK was mediated through the inhibition of mTOR, a highly conserved serine/threonine kinase that promotes protein synthesis, cell proliferation, and cell growth. The compound C, an AMPK inhibitor, and the mTOR activator MHY1485 reduce the anti-hypertrophic effect of GLP-1. The findings indicate that GLP-1 RA regulates the expression of Ang II/AT1R/ACE2 and activates the AMPK/mTOR pathway, thereby preventing cardiac hypertrophy [158], suggesting that GLP-1 agonists could be a useful treatment for patients with pathologic cardiac hypertrophy (Figure 3B).

In T2D patients, AMPK phosphorylation measured in PBMCs is significantly reduced [159]. In cultured human aortic endothelial cells, liraglutide, a GLP-1 RA, induces ADAM10-dependent ectodomain shedding of receptor advanced glycation end products (RAGE) via AMPK activation [160]. However, there are no clinical studies assessing the effect of GLP-1 RA on the activation of the cardioprotective AMPK/mTOR pathway in patients with metabolic syndrome. Thus, it is possible that patients with MS or T2D may have reduced AMPK phosphorylation levels (in PMBCs) compared to healthy people, and GLP-1 RA could increase AMPK phosphorylation levels.

In patients with type 2 diabetes (T2D) treated with glucose-lowering drugs, we observed a significant increase in Rho-associated protein kinase (ROCK) activation, as measured by two direct ROCK targets in peripheral blood mononuclear cells (PBMCs), by 4.2-fold and 5.7-fold (*p* < 0.001). Increased ROCK activation was significantly correlated with serum glucose levels [138]. Higher levels of ROCK activity have also been reported in patients with metabolic syndrome (MS) [161,162]. In HFrEF patients, a significant reduction in left atrial diameter was observed three months after effective resynchronization therapy, which was associated with a significant reduction of ROCK activation in circulating leukocytes [163]. Although GLP-1 RA treatment reduced ROCK activation and hypertrophy in cultured cardiac myoblasts stimulated by angiotensin II [155], there are no available human data on the effect of GLP-1 RA on ROCK activation in patients with MS or T2D, and GLP-1 RA may reduce ROCK activation in PBMCs, as well as body weight, plasma glucose, and insulin levels; increase aerobic capacity; and possibly decrease left atrial volume while improving left atrial function, potentially preventing HF in patients with MS or T2D.

In a mouse model resembling the cardiometabolic HFpEF phenotype, aged female mice were fed with a high-fat diet (HFD) and Ang II, and the GLP-1 RA liraglutide attenuated the cardiometabolic dysregulation and reduced cardiac hypertrophy, myocardial fibrosis, atrial weight, natriuretic peptide levels, and lung congestion [164]. In humans, a small clinical trial [165] examined the efficacy and safety of the GLP-1 analog liraglutide in the treatment of T2D patients on peritoneal dialysis (*n* = 16). The LV mass index (LVMI) was significantly reduced by 20% after 12 months with liraglutide [165]. In another study with 97 patients with T2D treated with liraglutide [166], blood glucose, blood pressure, albuminuria, and LVMI were significantly reduced after 24 months of treatment (*p* < 0.05) [166].

Interestingly, in the context of HFpEF, where exercise is a major therapeutic resource, it has been observed in an elegant preclinical model in mice that GLP-1 regulates skeletal muscle remodeling to enhance exercise endurance, possibly via GLP-1R signaling mediated phosphorylation of AMPK [167]. In the STEP-HFpEF Trial, which involved patients with HFpEF and obesity, exercise capacity was significantly improved in those receiving the GLP-1 receptor agonist semaglutide compared to those receiving a placebo over a period of 52 weeks [168]. Specifically, patients receiving semaglutide showed a 21.5 m improvement in the 6 min walking distance test, while those receiving the placebo only showed a 1.2 m improvement. Additionally, the semaglutide group experienced a 13.3% reduction in body weight compared to a 2.6% reduction in the placebo group [168]. In the abovementioned study, a similar change in exercise capacity was also observed in patients with a mildly reduced ejection fraction [169].

### 4.3. Current Evidence on Glucagon-like Peptide-1 Receptor Agonists (GLP-1 RAs) for Cardiovascular Prevention and Clinical Innovations

In a meta-nalysis including 800 patients with myocardial infarction undergoing angioplasty, GLP-1 RA vs. placebo treatment increased LVEF by 2.5% and reduced the infarct size by 5.3 g (95% CI: −10.4 to −0.2) [170]. GLP-1 RAs reduce the risk of CV events in T2D patients by different mechanisms. Other than a slight reduction of glycemic levels, GLP-1 RAs produce a significant weight loss, activate the cAMP-protein kinase A pathway, and inhibit RAS activation, exerting anti-hypertensive, antiatherosclerotic, anti-inflammatory, and reno-protective effects. GLP-1 RAs reduce the risk of myocardial infarction, CV death, and stroke [2,171]. Although GLP-1 RAs prevent atherosclerotic cardiovascular disease events [172], it is uncertain whether they prevent heart failure [171]. In a meta-analysis of more than 56,000 patients with T2DM, GLP-1 RAs showed statistically significant reductions in MACE (by 12%) and all-cause mortality (by 12%), as well as a broad renal composite outcome (by 17%) and a lower risk of hospitalization for HF (by 9%) [173].

In the recent STEP-HFpEF study in patients with HFpEF and obesity, treatment with the GLP-1 RA semaglutide for one year led to greater reductions in symptoms and physical limitations, greater improvements in exercise function, and greater weight loss than the placebo [17]. In addition, semaglutide reduced inflammation and body weight to a similar extent in all categories of LVEF [169]. In the meta-analysis by Avogaro et al. evaluating the effects of GLP-1 RAs on N-terminal pro-BNP, which included all randomized trials comparing GLP-1 RAs with the placebo (nine trials, 543 patients in GLP-1 RA, 536 in the placebo), a greater reduction in NT-proBNP levels was observed with GLP-1 RAs than with the placebo (−0.14 SD, *p* = 0.03), independent of baseline age, body weight, and metabolic control [174]. Thus, early treatment with GLP-1 RA may either mitigate or delay the risk of future HF in patients with T2D [174]. In a new meta-regression analysis in T2D patients (60,079 patients, 30,693 with GLP-1 RAs) reductions of HbA1C were associated with the reduction of three-point MACE-cardiovascular death, myocardial infarction, or stroke [174]. A three-point MACE (Log RR −0.290, *p* = 0.012) was observed, with an estimated RR reduction of 25% for each 1% absolute reduction in HbA1C levels [175]. Body weight loss was associated with the reduction of a three-point MACE (Log RR −0.068], *p* = 0.047), with an estimated RR reduction of 7% for each 1 kg reduction in body weight, while reductions of SBP and LDL-C were not associated with the reduction of three-point MACE [175]. Thus, early treatment with GLP-1 RA may either mitigate or delay the risk of future HF in patients with T2D [175].

In the SELECT multicenter, a double-blind, randomized, placebo-controlled trial with semaglutide was conducted on patients ≥ 45 years of age with preexisting cardiovascular disease and a body-mass index ≥ 27 but without diabetes (8803 were randomized to semaglutide and 8801 to the placebo, mean duration of follow-up 39.8 ± 9.4 months) [176]. A primary cardiovascular end-point event occurred in 6.5% of the semaglutide group and in 8.0% of the placebo group (hazard ratio, 0.80; 95% confidence interval, 0.72 to 0.90; *p* < 0.001), while adverse events leading to permanent discontinuation of the trial product occurred in 16.6% in the semaglutide group and in 8.2% in the placebo group (*p* < 0.001). In this study, weekly subcutaneous semaglutide was superior to the placebo in reducing the incidence of death from cardiovascular causes, nonfatal myocardial infarction, or nonfatal stroke [176].

Recently, a new generation of dual glucagon-like peptide-1 (GLP-1) and glucose-dependent insulin-stimulating polypeptide receptor agonists (GIPR) has been developed. The first of these is tirzetapide [177]. In eight randomized controlled trials involving 7491 patients with type 2 diabetes mellitus (T2D) [178], compared with GLP-1 RAs, insulin, and the placebo groups, body weight and blood pressure were reduced in the tirzetapide-treated groups of overweight/obese T2DM patients, which was dose-dependent. The GLP-1/GIP dual receptor agonists resulted in better weight loss than GLP-1 RA alone [178]. Furthermore, a novel triple receptor agonist peptide (retatrutide) that targets the glucagon receptor, the glucose-dependent insulinotropic polypeptide receptor (GIPR), and the GLP-1R has been developed with the potential to treat metabolic abnormalities associated with obesity as well as diseases resulting from it due to its distinct mechanism of action [179]. In people with T2D, in a phase II study (281 participants) with a placebo, GLP-1 RA dulaglutide, and retatrutide, the triple receptor agonist peptide showed clinically meaningful improvements in glycemic control and body weight reduction dose-dependently at 36 weeks by 3.2%, 10.4%, 16.8%, and 16.9% (with doses of 0.5, 4, 8, and 12 mg once a week, respectively) compared to 3.0% with the placebo and 2.0% with a safety profile consistent with GLP-1 receptor agonists and GIP and GLP-1 receptor agonists [180]. In this trial, 24.2% weight loss was observed at 48 weeks with 12 mg retatrutide [181]. This phase II trial provided informed dose selection for a phase 3 program.

With the current pathophysiological data, clinical evidence, and recent innovations about the relevant role of GLP-1 RA in preventing HFpEF and its effects on reverse remodeling and clinical outcomes, important mechanistic questions remain in patients with obesity, metabolic syndrome, or in obese type II diabetic patients. In the abovementioned clinical conditions, these questions can be broadly divided into two main groups. First, does GLP-1 RA significantly activate relevant cardioprotective/reverse remodeling mechanisms, such as the endogenous counter-regulatory RAS (ACE2 and angiotensins 1–7 and 1–9) and the AMPK/mTOR pathway? Secondly, can GLP-1 RA be considered a significant preventive tool for critical deleterious cardiac remodeling mechanisms, such as the PKA/RhoA/ROCK pathway, aldosterone effects, and systemic microinflammation consequences?

Administering GLP-1 RA to patients with MS or T2D may be a preventive measure for HFpEF. It can induce weight loss, improve glucose and insulin levels, increase aerobic capacity, and improve left atrial volume and function, as well as LV structure and function. This could be accomplished by increasing cardioprotective levels of Ang 1–9, 1–7, ACE 2, NEP, and the AMPK/mTOR pathway, while decreasing levels of Ang II, aldosterone, and Rho kinase activation (Table 2 and Figure 4), and could have implications for innovating more effective treatments for these conditions. However, it is important to note that further clinical and preclinical research is required to validate and enhance the clinical impact of these mechanisms.

## Figures and Tables

**Figure 1 ijms-25-04407-f001:**
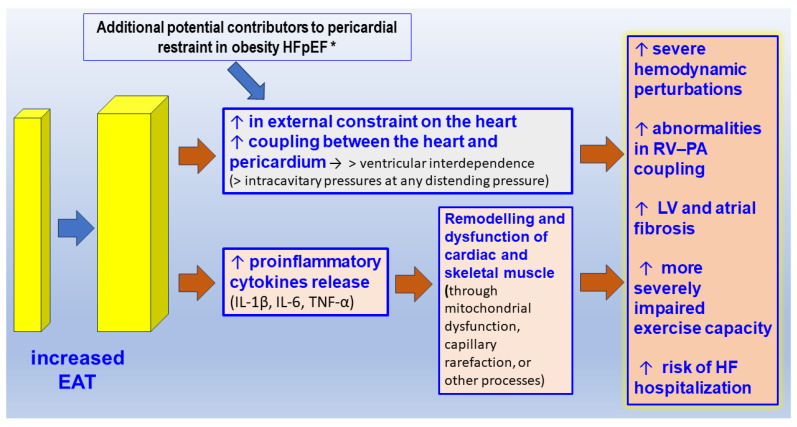
Mechanical and paracrine effects of increased epicardial adipose tissue (EAT) on myocardial structure and function (adapted from [102,105,106]). * = Additional potential contributors to pericardial restraint in obesity HFpEF: increased LV and RV volumes, increased left and right atrial volumes, blood volume expansion, mediastinal/abdominal extrinsic restraint, pulmonary hypertension with abnormal RV–PA coupling). Abbreviations: LV = left ventricle, RV = right ventricle, PA = pulmonary artery, IL = interleukin TNF = tumor necrosis factor, HF = heart failure.

**Figure 2 ijms-25-04407-f002:**
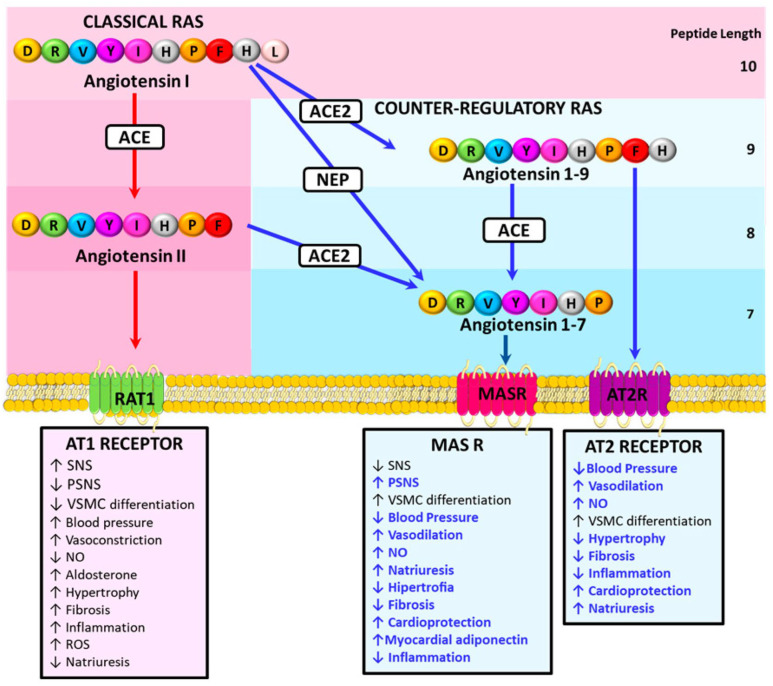
Preventive effects of the counter-regulatory renin angiotensin system (non-classic renin angiotensin pathway) on obese HFpEF. Activation of the AT2R and the MASR with angiotensin 1–9 and angiotensin 1–7, respectively, achieves protective effects on obese HFpEF by counteracting the classic effects of the RAS pathway activation. Additionally, GLP-1 RA also promotes activation of the counter-regulatory renin angiotensin system. Abbreviations: ACE = angiotensin converting enzyme 1, ACE2 = angiotensin converting enzyme 2, NEP = neutral endopeptidase, AT1R = angiotensin receptor type 1, AT2R = angiotensin receptor type 2, MASR = G-protein–coupled receptor Mas, SNS = sympathetic nervous system, PSNS = parasympathetic nervous system, VSMC = vascular smooth muscle cell.

**Figure 3 ijms-25-04407-f003:**
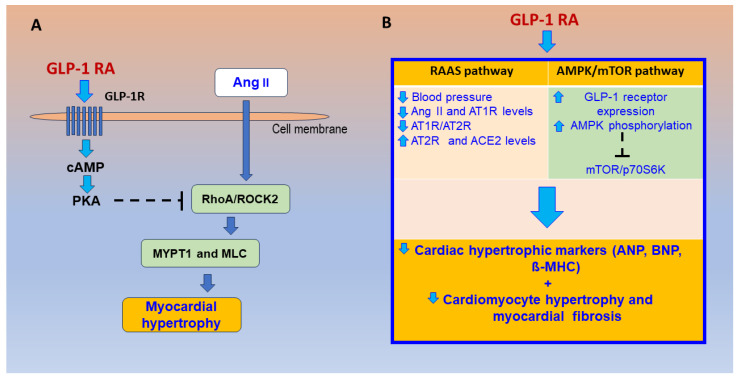
Mechanisms of GLP-1 RA on prevention of pathologic cardiac hypertrophy by (**A**) regulating the expression of the Ang II/AT1R/ACE2 pathway by inhibiting Ang II-induced RhoA/ROCK2 activation (which regulates MYPT1 and MLC phosphorylation) and (**B**) by activating the AMPK/mTOR/p70S6K pathway (adapted from Refs. [153,155]). Interrupted line = inhibition. Abbreviations: GLP-1 RA = glucagon-like peptide 1 receptor agonism, GLP-1R = glucagon-like peptide 1 receptor, cAMP = cyclic adenosine 3′,5′-monophosphate, PKA = protein kinase A, Ang II = angiotensin II, ROCK2 = Rho kinase 2, MYPT1 = Myosin phosphatase target subunit 1, MLC = myosin light chain, RAAS = renin angiotensin aldosterone system, AT1R = angiotensin receptor type 1, AT2R = angiotensin receptor type 2, ACE2 = angiotensin converting enzyme 2, AMPK = Adenosine monophosphate (AMP)-activated protein kinase, mTOR = mammalian target of rapamycin, p70S6K = p70 ribosomal S6 protein kinase, ANP = atrial natriuretic peptide, BNP = brain natriuretic peptide, β-MHC = β myosin heavy chain.

**Figure 4 ijms-25-04407-f004:**
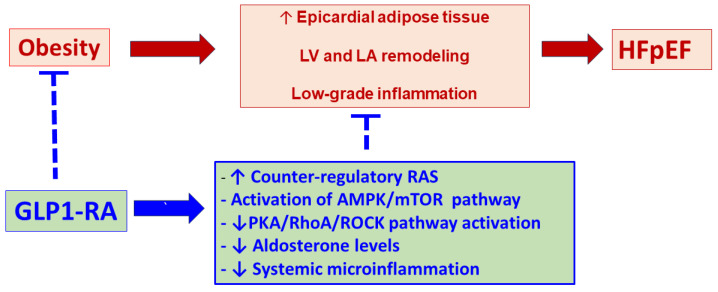
Mechanisms to prevent HFpEF using a glucagon-like peptide-1 receptor agonist (GLP-1 RA) in obesity (as well as in metabolic syndrome and in obese type 2 diabetes). The protective effects of GLP-1 on HFpEF may be attributed to the effects on obesity and its consequences on its ability to inhibit Ang II-induced RhoA/ROCK2 activation, which, in turn, regulates MYPT1 and MLC phos-phorylation, thus preventing pathologic myocardial remodeling and HFpEF. Interrupted line = in-hibition. Abbreviations: LV = left ventricle, LA = left atrium, RAS = renin angiotensin system, AMPK = AMP-activated protein kinase, mTOR = mammalian target of rapamycin, PKA = protein kinase A, ROCK = Rho-associated coiled-coil containing kinase.

**Table 1 ijms-25-04407-t001:** Pathophysiological mechanisms of HFpEF (modified from Ref. [2]).

	Major Comorbidities and Concomitant Conditions in HFpEF
-**Myocardial Remodeling**. Cardiomyocyte hypertrophy and changes in the composition and structure of the extracellular matrix with inflammatory cells and fibrillar collagen deposition (myocardial fibrosis).-**Endothelial oxidative stress** (with reduced nitric oxide bioavailability promoting vasoconstriction and pro-inflammatory and pro-thrombotic state). 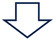 -**Impaired ventricular and atrial active relaxation and ****↑**** passive stiffness (diastolic dysfunction)**-**↓ Arterial compliance** (→ abnormal ventricular–arterial coupling and cardiac output reserve)	**Modifiable**	**Non-Modifiable**
**Hypertension****Obesity****Diabetes****Dyslipidemia****Metabolic syndrome**Renal dysfunctionAtrial fibrillationSleep apneaPulmonary HypertensionAmyloidosis	Advanced ageFemale sex

**Table 2 ijms-25-04407-t002:** Overview of key concepts and evidence for the prevention of HFpEF by glucagon-like peptide-1 receptor agonism (GLP-1 RA) in cardiometabolic diseases.

- The keystone of heart failure (HF) is left ventricular (LV) remodeling. - Of the major phenotypes of HFpEF, the obese or cardiometabolic phenotype is highly prevalent. - Obesity promotes increased pro-inflammatory cytokines and dysregulation of adipokines. - Epicardial adipose tissue (EAT) is a very active endocrine organ that secretes inflammatory cytokines, releases excessive fatty acids, and increases mechanical load on the myocardium resulting in myocardial remodeling - EAT may lead to increase myocardial fat content and interstitial fibrosis, which can reduce myocardial diastolic function and contractility. In humans, EAT thickness is reduced with the use of GLP-1 RA. - Exercise, diet, bariatric surgery, and pharmaceutical interventions reduce cardiac adipose tissue volume, and weight loss has a positive impact on the clinical course of HF. - The renin angiotensin system may have a significant role in regulating the expression of the Ang II/AT1R/ACE2 pathway on the cardioprotective effects of GLP-1 receptor agonism (preclinical observation). - GLP-1 RA may reverse myocardial hypertrophy through the PKA/RhoA/ROCK signaling pathway (preclinical observation). - GLP-1 RA regulates the expression of Ang II/AT1R/ACE2 and activates the AMPK/mTOR pathway, thereby preventing cardiac hypertrophy (preclinical observation). - A three-point MACE (cardiovascular death, myocardial infarction, or stroke) with GLP-1 RAs has been observed in T2D patients with an estimated risk reduction of 7% for each 1 kg reduction in body weight. - In patients with pre-existing cardiovascular disease, overweight, or obesity but without diabetes, the use of the GLP-1 RA semaglutide reduced primary cardiovascular end-point significantly by 20% - New generations of dual glucagon-like peptide-1 (GLP-1) + glucose-dependent insulin-stimulating polypeptide receptor agonists (GIPR) as well as triple receptor agonists targeting the glucagon receptor, GIPR, and the GLP-1R have been developed with promising clinical results in terms of body weight and blood pressure reduction

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
