# Peer review of "New Mechanisms to Prevent Heart Failure with Preserved Ejection Fraction Using Glucagon-like Peptide-1 Receptor Agonism (GLP-1 RA) in Metabolic Syndrome and in Type 2 Diabetes: A Review"

_ijms, 2024, doi:10.3390/ijms25084407_

Round 1

Reviewer 1 Report

Comments and Suggestions for Authors

Reviewer Report:

Manuscript Title: "New Mechanisms to Prevent Heart Failure with Preserved Ejection Fraction by Glucagon-Like Peptide-1 Receptor Agonism (GLP-1 RA) in Metabolic Syndrome and in Type 2 Diabetes: A Review"

General Comments: The manuscript presents a review of the mechanisms underlying the prevention of heart failure with preserved ejection fraction (HFpEF) by Glucagon-Like Peptide-1 Receptor Agonism (GLP-1 RA) in individuals with metabolic syndrome and type 2 diabetes. While the topic is of significant importance, the manuscript, in its current form, lacks sufficient novelty and requires major revisions before acceptance.

Specific Comments:

  1. Novelty: The manuscript falls short in offering novel insights or perspectives on the topic. The presented information seems to be largely derivative of existing literature without substantial new contributions to the field. To enhance the novelty of the manuscript, the authors should strive to incorporate recent advancements or emerging concepts related to GLP-1 RA and its mechanisms in preventing HFpEF.

  2. Figures: The manuscript lacks sufficient visual aids to complement the textual content. It is recommended that the authors include two to three additional high-definition figures illustrating the mechanistic pathways discussed in the manuscript. These figures will not only enhance the clarity of the presented mechanisms but also improve the overall readability and engagement of the review.

  3. Tables: To improve the clarity for readers, I suggest the inclusion of a comprehensive table summarizing all relevant data presented in the study. This table should provide a concise overview of key findings, mechanisms, and outcomes discussed throughout the manuscript. Such a table will facilitate better understanding and interpretation of the study's results.

  4. References: The manuscript would benefit from a more extensive review of recent literature. With only three references from 2023, the reference list appears outdated and may overlook critical studies published after that period. I recommend that the authors incorporate additional recent references to strengthen the review's credibility and relevance.

Overall, while the manuscript addresses an important topic, it requires substantial revisions to enhance its novelty, clarity, and comprehensiveness. I recommend that the authors address the aforementioned concerns before resubmitting the manuscript for further consideration.

Author Response

RESPONSES TO REVIEWER 1 

Reviewer Report: 1

Manuscript Title: "New Mechanisms to Prevent Heart Failure with Preserved Ejection Fraction by Glucagon-Like Peptide-1 Receptor Agonism (GLP-1 RA) in Metabolic Syndrome and in Type 2 Diabetes: A Review"

General Comments: The manuscript presents a review of the mechanisms underlying the prevention of heart failure with preserved ejection fraction (HFpEF) by Glucagon-Like Peptide-1 Receptor Agonism (GLP-1 RA) in individuals with metabolic syndrome and type 2 diabetes. While the topic is of significant importance, the manuscript, in its current form, lacks sufficient novelty and requires major revisions before acceptance.

Specific Comments:

  1. Novelty: The manuscript falls short in offering novel insights or perspectives on the topic. The presented information seems to be largely derivative of existing literature without substantial new contributions to the field. To enhance the novelty of the manuscript, the authors should strive to incorporate recent advancements or emerging concepts related to GLP-1 RA and its mechanisms in preventing HFpEF.

RESPONSE: Thanks very much for this comment.  In this revised version more recent discoveries and concepts on possible protective new mechanisms are discussed regarding HFpEF prevention by GLP1 AR.  In the current version a more comprehensive novel perspective has been taken in some aspects: data on the proposed protective mechanisms, new clinical studies providing contemporary evidence and new developments with this class of drugs in combination such as a dual GLP-1 and glucose-dependent insulin-stimulating polypeptide receptor agonists as well as the GLP-1/GIP/glucagon receptor triagonism.

  1. Figures: The manuscript lacks sufficient visual aids to complement the textual content. It is recommended that the authors include two to three additional high-definition figures illustrating the mechanistic pathways discussed in the manuscript. These figures will not only enhance the clarity of the presented mechanisms but also improve the overall readability and engagement of the review.

RESPONSE: Thanks for this observationAccording to this suggestion we have introduced 3 new Figures in order to illustrate the mechanistic pathways discussed in the paper. Specifically, 

Figure 1 address the mechanical and paracrine effects of increased epicardial adipose tissue (EAT) on myocardial structure and function

Figure 2  address the Counter-regulatory renin angiotensin system (non classic pathway renin angiotensin pathway) on obese HFpEF prevention

Figure 3  illustrates how GLP1 AR can prevent pathologic cardiac hypertrophy (and consequently of HFpEF) by regulating the expression of the AngII/AT1R/ACE2 pathway, by inhibiting AngII-induced RhoA/ROCK2 activation (which regulates MYPT1 and MLC phosphorylation)  and by activating the  AMPK/mTOR/p70S6K pathway

Former Figure 1 (a summary figure) is now Figure 4

  1. Tables: To improve the clarity for readers, I suggest the inclusion of a comprehensive table summarizing all relevant data presented in the study. This table should provide a concise overview of key findings, mechanisms, and outcomes discussed throughout the manuscript. Such a table will facilitate better understanding and interpretation of the study's results.

RESPONSEThanks again for this comment. The revised version includes a Table 2 (Overview of key concepts and evidence for the prevention of HFpEF by Glucagon-like Peptide-1 Receptor Agonism in cardiometabolic diseases)

Table 2. Overview of key concepts and evidence for the prevention of HFpEF by Glucagon-like Peptide-1 Receptor Agonism (GLP-1 RA) in cardiometabolic diseases

The keystone of heart failure (HF) is left ventricular (LV) remodeling

Of the major phenotypes of HFpEF, the obese or cardiometabolic phenotype is highly prevalent.

Obesity promotes increased pro-inflammatory cytokines and dysregulation of adipokines.

Epicardial adipose tissue (EAT) is a very active endocrine organ that secretes inflammatory cytokines, releases excessive fatty acids, and increases mechanical load on the myocardium resulting in myocardial remodeling

EAT may lead to increase myocardial fat content and interstitial fibrosis, which can reduce myocardial diastolic function and contractility. of In humans, EAT thickness is reduced with the use of GLP1 AR.

Exercise, diet, bariatric surgery and pharmaceutical interventions reduce cardiac adipose tissue volume and weight loss has a positive impact on the clinical course of HF

The renin angiotensin system may have a significant role in regulating the expression of the AngII/AT1R/ACE2 pathway on the cardioprotective effects of GLP1 receptor agonism (preclinical observation)

GLP1 RA may reverse myocardial hypertrophy through the PKA/RhoA/ROCK signaling pathway (preclinical observation)

GLP1 RA regulates the expression of AngII/AT1R/ACE2 and activates the AMPK/mTOR pathway, thereby preventing cardiac hypertrophy (preclinical observation)

3-point MACE (cardiovascular death, myocardial infarction, or stroke) with GLP-1 RAs has been observed in T2D patients with an estimated risk reduction of 7% for each 1 kg reduction in body weight

In patients with preexisting cardiovascular disease, with overweight or obesity but without diabetes the use of the GLP1 AR semaglutide reduced primary cardiovascular end-point significaly by 20%

New generations of dual glucagon-like peptide-1 (GLP-1) + glucose-dependent insulin-stimulating polypeptide receptor agonists (GIPR) as well as triple receptor agonists targeting  the glucagon receptor, GIPR, and the GLP-1R have been developed with promising clinical results in terms of body weight and blood pressure reduction

  1. References: The manuscript would benefit from a more extensive review of recent literature. With only three references from 2023, the reference list appears outdated and may overlook critical studies published after that period. I recommend that the authors incorporate additional recent references to strengthen the review's credibility and relevance.

RESPONSEFollowing this very constructive observation - and in accordance with comment 1 on novelty - new relevant references have been included in the current version (twenty one from  year 2023 and sixteen from year 2024)

Overall, while the manuscript addresses an important topic, it requires substantial revisions to enhance its novelty, clarity, and comprehensiveness. I recommend that the authors address the aforementioned concerns before resubmitting the manuscript for further consideration.

Reviewer 2 Report

Comments and Suggestions for Authors

First of all, I want to thank the authors for their great contribution to the literature on diabetes, HFpEF, and obesity. Following are my comments and I do suggest some major revisions prior to publication of the paper:

Title

The first 2/3 of the review largely focus on explaining very detailed molecular and pathophysiological processes behind obesity, T2DM, and HFpEF with GLP agonism having somewhat a secondary objective in the current form. Authors could potentially consider revising the title of the paper to reflect the association between described conditions rather than emphasize preventative role of GLP-1. However, I leave it to authors discretion to decide on the most suitable title.

Introduction

It appears that only introduction as a section was included in the manuscript, there are no subsections of materials & methods, discussion, and conclusion. Furthermore, please avoid writing subsections in the introduction, which should have only one section. I would recommend either adding other appropriate sections or removing the “Introduction” all together.

Line 35: Please include the full description for the abbreviation as this is the first time it is being reported in the text (not including the titles/subtitles). Please rewrite this statement, did authors mean to say “is a form of heart failure present in 50% of patients diagnosed with HF?”

Line 36: Add years of age here and throughout the manuscript where appropriate.

Table 1:

Consider dividing second column of the table into two subsections to include factors which are modifiable (e.g. HTN, obesity, DM) and those which are not (age, sex). For the first column, I would suggest presenting the mechanisms chronologically first with the process of remodeling and oxidative stress leading to impaired relaxation and stiffness resulting in diastolic dysfunction, impaired coupling etc. Also, I am not sure if a single cardio myocyte can be called stiff, it does make sense when referring to the ventricle but not to a single cell.

Line 43: I would advise removing this sentence: “Without LV remodeling..”

Line 45: “replacement of myocardium with fibrosis”

Line 50: Please mention which phenotypes these are.

Line 57: Delete this part of the sentence: “and cardiac remodeling plays...”

Line 58-60: Please revise this sentence, consider deleting the first sentence and writing the global prevalence instead of prevalence only in Chile.

Line 61: State what the correlation is,  i.e. the prevalence increases with age.

Line 61/62: Please add references to this statement.

Line 79: Please add references

Line 81: This sentence should be presented more clearly – do authors imply that EAT leads to these conditions or these conditions increase the incidence of EAT.

Line 84-94: I think this is not relevant to the topic of the paper, consider deleting.

Line 95: Sentence can be rewritten a bit more clearly; which disorders are in question?

Line 97: delete word heart

Paragraph 98-107: This belongs in the “Current treatment..” subsection

Line 109: Please expand further on how they exhibit both a pro and anti-inflammatory effect?

Line 114: “this population”

Line 115-119: This paragraph needs to be placed into context a bit. It appears that authors suggest that higher leptin and lower adiponectin are encountered in obesity, however following weight reduction surgery these hormone abnormalities are even more pronounced? Is there an explanation for this?

Line 121, 124: Please provide references.

Line 161: Please provide explanation behind why hypoxia occurs.

Line 165: What does further formation of inflammasome lead to? Please explain the complete pathophysiology.

Line 166: Is this process only a risk factor in those who already have heart disease or all otherwise healthy obese individuals? I would rephrase to “worsens cardiovascular function” if the second is correct.

Line 170: Please comment further on the pathophysiology of how this collagen deposition influences relaxation of the LV

Paragraph 2.2: This passage in general is repetitive to what was explained above and I would advise authors to try to combine this with the previous sections.

Line 190: delete “when obesity worsens”

Line 197: “Increase in subcutaneous AT was not associated with increase in ***” – Please write whether authors referred to HFpEF hospitalizations, mortality, etc.

Line 200: Increase in visceral AT?

Line 228: Sentence should be rewritten to describe at least some of the direct effects as an introduction to the passage.

Line 237: Please add “imaging”

Paragraph 2.6: This section is well written and information is significant, however I would suggest breaking it up and including in the passages where the therapeutic options are discussed - 3. & 4.

Line 300-311: Overall I am not sure if this passage adds any value to the text. Studies are limited, and I am not entirely sure how we can interpret this and put it into clinical context. Furthermore, authors talk about HFrEF throughout the second half of the passage which is not the focus of this review. I would advise authors to revise this passage, either make it shorter and more succinct or deleting it all together.

Line 300/301: BMI, HbA1C - What do these values refer to? If this is a mean value noted in the study in question, this needs to be explained in the paper.

Line 313-316: Similar to previous comment, I am not sure if this paragraph adds value to the topic of this review.

Line 317-325: I would advise authors to rewrite this paragraph to provide more clarity.

Line 332-335: I would advise authors to move this paragraph to section number 4.

Line 353: I would advise deleting this as it was mentioned previously.  

Line 359: I advise rewriting this sentence since it makes it sound like LV remodeling causes T2D.

Line 366: except instead of excepting

Line 367: pronounces instead of marked

Line 477-479: This sentence is grammatically incorrect, please rewrite.

Paragraph 4 – Overall I think that this paragraph is very well written, provides a significant content of information both in terms of basic science and clinical relevance.

In conclusion, I think that authors wrote a good review and I enjoyed reading it. It is informative and asks important questions.

Comments on the Quality of English Language

Minor editing is required. 

Author Response

RESPONSES TO REVIEWER 2

Reviewer Report: 2

First of all, I want to thank the authors for their great contribution to the literature on diabetes, HFpEF, and obesity. Following are my comments and I do suggest some major revisions prior to publication of the paper:

Title

The first 2/3 of the review largely focus on explaining very detailed molecular and pathophysiological processes behind obesity, T2DM, and HFpEF with GLP agonism having somewhat a secondary objective in the current form. Authors could potentially consider revising the title of the paper to reflect the association between described conditions rather than emphasize preventative role of GLP-1. However, I leave it to authors discretion to decide on the most suitable title.

RESPONSE: Thanks for this gentle observation. In the current version a more comprehensive perspective has been taken in aspects related with a) proposed protective mechanisms of HFpEF by GLP1 AR, b) new clinical studies providing contemporary good evidence with GLP1 AR in cardiometabolic prevention and in HFpEF and c) additionally, new developments with this class of drugs in combinations such as the dual GLP-1 AR + glucose-dependent insulin-stimulating polypeptide (GIP) receptor agonists as well as the GLP-1/GIP/glucagon receptor triagonism. In line with these changes and the reviewers' comment, the original title was kept.

Introduction

It appears that only introduction as a section was included in the manuscript, there are no subsections of materials & methods, discussion, and conclusion. Furthermore, please avoid writing subsections in the introduction, which should have only one section. I would recommend either adding other appropriate sections or removing the “Introduction” all together.

RESPONSE: According to this very useful suggestion we have preferred to remove the word Introduction and at the same time to emphasize at the end  the 3 main aspects that will be analyzed in the following sections :

  1. Consequences of Obesity on the Heart and on Heart Failure with Preserved Ejection Fraction
  2. The Counter-Regulatory Renin Angiotensin System and Prevention of Cardiovascular Damage
  3. The Glucagon-Like Peptide 1 (GLP-1) Axis: A Novel Anti Diabetic, Anti Obesity and Anti Cardiac Remodeling Path

Line 35: Please include the full description for the abbreviation as this is the first time it is being reported in the text (not including the titles/subtitles). Please rewrite this statement, did authors mean to say “is a form of heart failure present in 50% of patients diagnosed with HF?”

RESPONSE: According to this observation the full description for the abbreviation was added and the whole sentence was changed to:

Heart failure with preserved ejection fraction (HFpEF) is a growing epidemic, accounting for 50% of all heart failure patients and the leading cause of hospitalizations in patients over 65 years of age

Line 36: Add years of age here and throughout the manuscript where appropriate.

RESPONSE: Thanks, this was modified accordingly

Table 1:

Consider dividing second column of the table into two subsections to include factors which are modifiable (e.g. HTN, obesity, DM) and those which are not (age, sex). For the first column, I would suggest presenting the mechanisms chronologically first with the process of remodeling and oxidative stress leading to impaired relaxation and stiffness resulting in diastolic dysfunction, impaired coupling etc. Also, I am not sure if a single cardio myocyte can be called stiff, it does make sense when referring to the ventricle but not to a single cell.

RESPONSE: Thanks for this very useful comment. Table 1 has been changed accordingly and the reviewer’s observations were incorporated to:

Table 1.  Pathophysiological mechanisms of HFpEF  (modified from ref 2)

Major comorbidities and concomitant

conditions in HFpEF

- Myocardial Remodeling. Cardiomyocyte hypertrophy and changes in the composition and structure of the extracellular matrix with inflammatory cells and fibrillar collagen deposition (myocardial fibrosis)

- Endothelial oxidative stress (with reduced nitric oxide bioavailability promoting vasoconstriction and pro-inflammatory and pro-thrombotic state).

of maladaptive cardiac remodeling

- ↆ arterial compliance (→ abnormal ventricular-arterial coupling and cardiac output reserve)

- Impaired ventricular and atrial active relaxation and ↑ passive stiffness (diastolic dysfunction)

Modifiable

Non Modifiable

hypertension

obesity

diabetes

dyslipidemia

metabolic syndrome

renal dysfunction

atrial fibrillation

sleep apnea

pulmonary hypertension

amyloidosis

advanced age

female sex

Line 43: I would advise removing this sentence: “Without LV remodeling..”

RESPONSE: According to this observation, this sentence was deleted 

Line 45: “replacement of myocardium with fibrosis” .

RESPONSE: According to this observation it was changed to leads to eccentric hypertrophy and myocardial fibrosis

Line 50: Please mention which phenotypes these are.

RESPONSE: Thanks for this observation. In the current version  the main current phenotypes are mentioned as

Distinct clinical HFpEF phenotypes are recognized. There are 6 major phenotypes of HFpEF described, characterized by distinct clinical features:  the aging phenotype; the obesity or cardiometabolic phenotype, the phenotype associated with arterial hypertension, the pulmonary arterial hypertension phenotype, the coronary artery disease phenotype, and the phenotype associated with left atrial myopathy [3]. All phenotypes of HFpEF have a variety of comorbidities in common. It is uncommon for HFpEF patients to have only one comorbid condition. The aging phenotype is often associated with comorbidities such as atrial fibrillation, anemia, chronic obstructive pulmonary disease (COPD), and frailty. On the other hand, the obesity phenotype is more commonly associated with comorbidities such as OSA, diabetes, and chronic kidney disease (CKD) [3].

Line 57: Delete this part of the sentence: “and cardiac remodeling plays...”

RESPONSE:  We thank the reviewer for this comment for further reflection on the manuscript subject. We have decided to keep this concept as it central in understanding the pathophysiology of HFpEF in metabolic syndrome and T2D. Besides, the pathophysiologic mechanisms discussed here (the endogenous counterregulatory RAS, the AMPK/mTOR pathway, the PKA/RhoA/ROCK pathway, aldosterone and systemic microinflammation)  have significant effects on cardiac remodeling and on HFpEF development as well as they are modulated by GLP1 RA. We hope this explanation be helpful in understanding this perspective.

Line 58-60: Please revise this sentence, consider deleting the first sentence and writing the global prevalence instead of prevalence only in Chile.

RESPONSE:  Thanks for this comment. In the current version, the global data, instead of local data are provided

The World Health Organization reports that 1.28 billion adults aged 30-79 worldwide have hypertension [8]. In 2016, 39% of adults aged 18 and over were overweight, and 13% were obese [8]. Additionally, approximately 422 million people worldwide have diabetes [8].. The prevalence of metabolic syndrome varies globally, ranging from 12.5% to 31.4%, depending on the definition used [9].

Line 61: State what the correlation is,  i.e. the prevalence increases with age

RESPONSE:  In the current version, and according to this observation substantial information regarding  age dependent prevalences were added.

The prevalence of these four conditions depends on age [10-13]. As we age, both overweight and obesity become more common, reaching their highest point between the ages of 50 to 65 years, and then showing a slight downward trend [11. It's worth noting that the prevalence of hypertension in adults in the United States (2017–2018) increases with age: 22.4% (aged 18–39), 54.5% (40–59), and 74.5% (60 and over) [10]. These high prevalences are associated with a high incidence of HFpEF [14-16]. To effectively prevent this, a recent randomized trial with a GLP1 RA in already obese HFpEF patients suggests taking action. The trial demonstrated larger reductions in symptoms and physical limitations, greater improvements in exercise function, and more significant weight loss than the placebo [17].

Line 61/62: Please add references to this statement.

RESPONSE:  In the current version, we added one reference for each condition

Line 79: Please add references

RESPONSE:  As suggested, in the current version, we added at least one reference for each condition

Line 81: This sentence should be presented more clearly – do authors imply that EAT leads to these conditions or these conditions increase the incidence of EAT.

RESPONSE: We agree completely with the reviewer's concern. However, in this context and with the available evidence, this sentence does not imply a causal relationship, which is not uncommon in medicine. Instead, it suggests a direct relationship that requires further investigation in order to determine causality.

Line 84-94: I think this is not relevant to the topic of the paper, consider deleting. 95  a 104

RESPONSE: According to the reviewer observation this paragraph was shortened by 60%. In this context, we consider it relevant to mention the relationship between EAT size and inflammation, and the possibility that EAT could transmit this inflammation to the heart.

Line 95: Sentence can be rewritten a bit more clearly; which disorders are in question?

RESPONSE: According to the reviewer’s observation this sentence was rewritten to

Each metabolic disorder that is linked to both atrial fibrillation (AF) and HFpEF is also accompanied by an expansion of epicardial adipose tissue (EAT) mass.[13]

Line 97: delete word heart 109.

RESPONSEin the current version the word heart was deleted

Paragraph 98-107: This belongs in the “Current treatment..” subsection. .

RESPONSEAccording to this observation this paragraph was moved to the “Current treatment..” subsection  and  additional references have been provided there

Line 109: Please expand further on how they exhibit both a pro and anti-inflammatory effect?

RESPONSE:  in the current version the pro and anti-inflammatory effects of leptin and adiponectin, respectively have been further expanded:

Leptin and adiponectin are two adipokines that can modulate insulin sensitivity and have been shown to have pro-inflammatory and anti-inflammatory effects, respectively [15]. Obesity is associated with increased leptin levels that activate the cells of the innate and adaptive immune system [32]. Leptin acts as an acute-phase reactant, increasing the secretion of proinflammatory cytokines such as IL-6, IL-12, and TNF-α in macrophages [33]. This, in turn, increases leptin expression in adipose tissue and circulating leptin, creating a feedback loop which promotes the inflammatory state. Obese individuals exhibit a greater abundance of macrophages in their adipose tissue, indicating that both adipocytes and macrophages play significant roles in inflammation in obesity [33]. High levels of leptin in obesity are associated with cardiac and renal fibrosis [34], increased aldosterone production, and sodium retention [35].

On the other side, adiponectin, which is exclusively produced by adipocytes and circulates in plasma, has a significant insulin-sensitizing effect in humans [16] and exerts anti-inflammatory effects especially in cardiometabolic diseases [36]. Serum adiponectin levels are inversely correlated with obesity and inflammatory cytokine levels in T2D and proinflammatory cytokines reduce adiponectin expression in adipocytes [36]. In vitro treatment of macrophages with adiponectin inhibits the pro-inflammatory cytokine TNF-a and induces the expression of the anti-inflammatory cytokine IL-10 [37]. In addition, adiponectin induces the M1 to M2 macrophage polarization switch, thereby attenuating chronic inflammation [36]. Low adiponectin levels in obesity increase the risk for cardiovascular disease [35] whereas adiponectin levels are reduced in HFpEF. In vitro, adiponectin has several protective effects: it stimulates AMP-activated protein kinase (AMPK)-dependent and extracellular signal-regulated kinase signalling in cardiac myocytes and endothelial cells [38], reduces hypertrophy and fibrosis, activates the endothelial nitric oxide synthase system and increases nitric oxide production [39,40].

Line 114: “this population” .

RESPONSEin the current version this was changed accordingly

Line 115-119: This paragraph needs to be placed into context a bit. It appears that authors suggest that higher leptin and lower adiponectin are encountered in obesity, however following weight reduction surgery these hormone abnormalities are even more pronounced? Is there an explanation for this?

RESPONSE: We appreciate the reviewer's comment. In the current version, we have provided context for this paragraph and made necessary corrections.

Line 121, 124: Please provide references.

RESPONSEin the current version references in this paragraph have been provided

Observational and community-based cohorts have reported an association between obesity and LV concentric remodeling [44,45]. Evidence of the central role of obesity in the pathogenesis of LV concentric remodeling is further supported by the correlation between weight loss and reduction of LV mass [46,47] and also between weight loss, LVH regression and blood pressure reduction after bariatric surgery. [48-50]. Obesity can impair LV diastolic function through mechanisms other than increased cardiac pre- and afterload [1] and is a known risk factor for incident HFpEF [34,51,52].

Line 161: Please provide explanation behind why hypoxia occurs.

RESPONSE. Thanks for this comment. According to this observation, in the current version we explain this as:

In addition, when nutrient availability is excessive, adipocytes store various lipid species in lipid droplets that rapidly increase in size and reach the diffusion limit of oxygen [74]. Thus, hypoxia is an important driver of the early AT response to increased dietary lipids [74]. Adipocyte hypoxia is induced as early as 1 and 3 days of a high fat diet [75]. Hypoxia during early AT expansion induces stress signaling that facilitates angiogenesis, initiates inflammatory cell infiltration, and induces organized ECM remodeling to encourage appropriate AT expansion. [74]. Adipogenesis affects the process of adipose tissue remodeling, while hypoxia triggers angiogenesis, extracellular matrix remodeling, and inflammation [74-76].

Line 165: What does further formation of inflammasome lead to? Please explain the complete pathophysiology.

RESPONSEThanks for this comment. In the current version, the inflammatory pathophysiology after activation of the inflammasome is explained as:

Inflammatory visceral adipose tissue produces reactive oxygen species and low levels of nitric oxide, leading to mitochondrial dysfunction and activation of the Nod-like receptor protein 3 inflammasome [77]. which serves as a platform for caspase-1 activation leading to the processing of proinflammatory cytokines IL-1ß, IL-18 and gasdermin D (GSDMD) mediated cell death [78]. This form of cell death represents a major pathway of inflammation [78].

Line 166: Is this process only a risk factor in those who already have heart disease or all otherwise healthy obese individuals? I would rephrase to “worsens cardiovascular function” if the second is correct.

RESPONSE. Thanks, in the current version this was changed as suggested

Line 170: Please comment further on the pathophysiology of how this collagen deposition influences relaxation of the LV

RESPONSE  In the current version we incorporated this issue as:

Increased myocardial collagen (myocardial fibrosis) increases left ventricular stiffness and filling pressure, leading to diastolic dysfunction. This makes passive inflow difficult and causes backward diastolic flow, further impairing early filling [85]. In this regard, there is direct relationship between the level  of  myocardial collagen, LV stiffness and pulmonary capillary wedge pressure [86]. 

Paragraph 2.2: This passage in general is repetitive to what was explained above and I would advise authors to try to combine this with the previous sections.

RESPONSE  Thanks, in the current version we incorporated this observation and combined with the previous sections. In consequence, there is no longer a paragraph named Adipocyte Dysfunction and HfpEF 

Line 190: delete “when obesity worsens” 

RESPONSE: According to the observation this was deleted

Line 197: “Increase in subcutaneous AT was not associated with increase in ***” – Please write whether authors referred to HFpEF hospitalizations, mortality, etc.

RESPONSE According to this observation this sentence was modified in order to state it clearly to:  Incident HFpEF hospitalizations in the population of the Jackson Heart Study were significantly associated with both visceral and epicardial AT, but not with subcutaneous AT.

Line 200: Increase in visceral AT?

RESPONSE: Yes, this was modified accordingly

Line 228: Sentence should be rewritten to describe at least some of the direct effects as an introduction to the passage.

RESPONSE: In the current version this sentence was rewritten to describe some of the direct effects as an introduction  as:

In addition to T2D macrovascular complications, T2D has direct effects on the myocardium such as left ventricular hypertrophy [109,110] a complex myocardial dysfunction, referred as diabetic cardiomyopathy (which  results in abnormal diastolic and systolic function) [111, 112], heart failure [113-115]  and also prolonged ventricular repolarization [116].

Line 237: Please add “imaging”.

RESPONSE: thanks, “imaging”was here added

Paragraph 2.6: This section is well written and information is significant, however I would suggest breaking it up and including in the passages where the therapeutic options are discussed - 3. & 4.

RESPONSE: Thanks very much for this comment. We understand the reviewer’s point, however section 2 covers consequences of Obesity on the Heart and on HFpEF finishing with what is currently known about evidence based effective treatment for HFpEF. Section 3 analyses the current knowledge on the parallel RAS (The Counter-Regulatory Renin Angiotensin System) in relation to the potential prevention of cardiovascular damage. However, there is currently no robust evidence-based data regarding effective treatment for HFpEF by stimulating the parallel RAS. On the other side section 4 is focused on the Glucagon-Like Peptide 1 (GLP-1) axis and analyzes its role as a novel anti diabetic, anti obesity and anti cardiac remodeling path with a high potential for effective HFpEF prevention.

Line 300-311: Overall I am not sure if this passage adds any value to the text. Studies are limited, and I am not entirely sure how we can interpret this and put it into clinical context. Furthermore, authors talk about HFrEF throughout the second half of the passage which is not the focus of this review. I would advise authors to revise this passage, either make it shorter and more succinct or deleting it all together.

and Line 313-316: Similar to previous comment, I am not sure if this paragraph adds value to the topic of this review.

RESPONSE TO BOTH OBSERVATIONS: Thanks and according to the reviewer observations, in the current version the 2 paragraphs have been revised and significantly shortened by 50%. The concept of the protective role of the counter-regulatory RAS, supported by limited human data, has been upheld.

The counter-regulatory RAS exhibits an inverse relationship between Ang II with Ang 1-9 , Ang 1-7 and ACE2 levels [86]. However, there is limited human data available on the levels of cardioprotective Ang 1-9 and ACE2 in MS, T2D, and heart failure. Plasma Ang II levels were 29% higher in stable T2D patients compared to controls (< 0.05) [89].   In a small study of patients with heart failure, plasma Ang 1-9 levels were 86% lower and Ang II levels were 30% higher compared to controls (p<0.05). In a small study of 11 patients with CKD and T2D, plasma levels of Ang 1-7 increased by 2.3-fold after 12 weeks of empagliflozin treatment [91].  No other data are currently available on Ang 1-9 levels in humans or on the effect of GLP1 receptor agonism on Ang 1-7, 1-9, ACE 2 or aldosterone levels in patients with MS or T2D in relation to cardiac phenotype protection.

Line 300/301: BMI, HbA1C - What do these values refer to? If this is a mean value noted in the study in question, this needs to be explained in the paper.

RESPONSE:  (This is related to  the Reviewer Observation for Lines 300-311) In the current version this passage with those data has been revised and significantly shortened, and these numbers were deleted

Line 317-325: I would advise authors to rewrite this paragraph to provide more clarity.

RESPONSE:  Thanks, acccording to this observation this paragraph was rewritten in order to provide more clarity to:

A meta-analysis conducted recently found that lower adipose tissue ACE2 expression in humans, specifically in abdominal subcutaneous fat, was associated with multiple adverse cardio-metabolic health indices. These included T2D, obesity status, increased serum fasting insulin levels, BMI, and lower serum HDL levels [142]. Specifically, individuals with cardio-metabolic features were found to have lower ACE2 expression and a reduced proportion of microvascular endothelial cells, but a high proportion of macrophages [142]. Thus, decreased ACE2 expression in adipose tissue may play a role in the development of cardio-metabolic disorders as well as in the increased risk of severe COVID-19.

Line 332-335: I would advise authors to move this paragraph to section number 4.

RESPONSE:  Thanks for this comment.  Accordingly,  this paragraph was moved  to the final paragraph :

 With the current pathophysiological  data, clinical evidence and  recent  innovations about the relevant role of GLP-1 RA in preventing HFpEF and its effects on reverse remodelling and clinical outcomes, important mechanistic questions remain in patients with obesity, metabolic syndrome or in obese type II diabetic patients. In the abovementioned clinical conditions these questions can be broadly divided into 2 main groups. First: does GLP-1 RA significantly activate relevant cardioprotective/reverse remodelling mechanisms such as the endogenous counterregulatory RAS (ACE2 and angiotensins 1-7 and 1-9) and the AMPK/mTOR pathway?  Secondly, can GLP-1 RA be considered a significant preventive tool for critical deleterious cardiac remodeling mechanisms, such as the PKA/RhoA/ROCK pathway, aldosterone effects, and systemic microinflammation consequences?.

Administering GLP-1 RA to patients with MS or T2D may be a preventive measure for HFpEF. It can induce weight loss, improve glucose and insulin levels, increase aerobic capacity, and improve left atrial volume and function, as well as LV structure and function. This could be accomplished by increasing cardioprotective levels of Ang 1-9, 1-7, ACE 2, NEP and the AMPK/mTOR pathway, while decreasing levels of Ang II, aldosterone and Rho kinase activation (Table 2 and Figure 4), and could have implications for innovating more effective treatments for these conditions. However, it is important to note that further clinical and preclinical research is required to validate and enhance the clinical impact of these mechanisms.

Line 353: I would advise deleting this as it was mentioned previously.  

RESPONSEthis was changed as suggested

Line 359: I advise rewriting this sentence since it makes it sound like LV remodeling causes T2D.

RESPONSE:  According to this observation this sentence was modified to:  Since in obese HFpEF, visceral and epicardial AT drive LV remodeling, aggressive weight management will benefit  HFpEF patients with obesity or T2D

Line 366: except instead of excepting

RESPONSEthis was changed as suggested

Line 367: pronounced instead of marked

RESPONSE:  this was changed as suggested

Line 477-479: This sentence is grammatically incorrect, please rewrite.

RESPONSEThanks for this observation, this sentence was changed to:

Secondly, can GLP-1 RA be considered a significant preventive tool for critical deleterious cardiac remodeling mechanisms, such as the PKA/RhoA/ROCK pathway, aldosterone effects, and systemic microinflammation consequences?

Paragraph 4 – Overall I think that this paragraph is very well written, provides a significant content of information both in terms of basic science and clinical relevance.

In conclusion, I think that authors wrote a good review and I enjoyed reading it. It is informative and asks important questions.

RESPONSE:  We are very grateful for the reviewer comments
